# APE1/Ref-1 as a Therapeutic Target for Inflammatory Bowel Disease

**DOI:** 10.3390/biom13111569

**Published:** 2023-10-24

**Authors:** Lauren Sahakian, Ainsley M. Robinson, Linda Sahakian, Rhian Stavely, Mark R. Kelley, Kulmira Nurgali

**Affiliations:** 1Institute for Health & Sport, Victoria University, Melbourne, VIC 3021, Australia; lauren.sahakian@live.vu.edu.au (L.S.); ainsley.robinson@vu.edu.au (A.M.R.); 2Department of Medicine Western Health, The University of Melbourne, Melbourne, VIC 3010, Australia; lindasahakian@hotmail.com (L.S.); rstavely@mgh.harvard.edu (R.S.); 3Department of Pediatric Surgery, Massachusetts General Hospital, Harvard Medical School, Boston, MA 02114, USA; 4Department of Pediatrics, Herman B Wells Center for Pediatric Research, Indiana University School of Medicine, Indianapolis, IN 46202, USA; mkelley@iu.edu; 5Department of Pharmacology and Toxicology, Indiana University School of Medicine, Indianapolis, IN 46202, USA; 6Regenerative Medicine and Stem Cells Program, Australian Institute for Musculoskeletal Science (AIMSS), Melbourne, VIC 3021, Australia

**Keywords:** apurinic/apyrimidinic endonuclease 1/reduction-oxidation factor 1 (APE1/Ref-1), redox signaling, inflammatory bowel disease (IBD), inflammation, oxidative stress

## Abstract

Inflammatory bowel disease (IBD) is characterized by chronic relapsing inflammation of the gastrointestinal tract. The prevalence of IBD is increasing with approximately 4.9 million cases reported worldwide. Current therapies are limited due to the severity of side effects and long-term toxicity, therefore, the development of novel IBD treatments is necessitated. Recent findings support apurinic/apyrimidinic endonuclease 1/reduction-oxidation factor 1 (APE1/Ref-1) as a target in many pathological conditions, including inflammatory diseases, where APE1/Ref-1 regulation of crucial transcription factors impacts significant pathways. Thus, a potential target for a novel IBD therapy is the redox activity of the multifunctional protein APE1/Ref-1. This review elaborates on the status of conventional IBD treatments, the role of an APE1/Ref-1 in intestinal inflammation, and the potential of a small molecule inhibitor of APE1/Ref-1 redox activity to modulate inflammation, oxidative stress response, and enteric neuronal damage in IBD.

## 1. Introduction

Inflammatory bowel disease (IBD) collectively describes two debilitating conditions, ulcerative colitis (UC) and Crohn’s disease (CD), characterized by chronic, relapsing, and remitting inflammation within the gastrointestinal (GI) tract [1]. While CD and UC are distinguishable by disease location within the GI tract, the nature of histopathological alterations in the intestinal wall, and associated complications, there is some overlap in clinical and pathological manifestations, such as GI disruptions, damage to the enteric nervous system (ENS), and modifications in immune responses [2]. UC is distinguished by diffuse continuous inflammation confined to the rectal and colonic mucosa and submucosa. Typically, the inflammatory reaction initiates within the mucosal and submucosal lining of the rectum, impacting either the rectum exclusively or extending proximally to involve all or part of the colon [3]. In CD, inflammation is segmented, transmural, and focal, most commonly manifesting within the colon and terminal ileum, but may affect any part of the GI tract [4]. Clinical symptoms of IBD include diarrhea, weight loss, fatigue, abdominal pain, perianal fissures, bloody stool, and tenesmus [5,6]. Disease progression may lead to complications including fistulas, strictures, abscesses (CD), rectal bleeding (UC), and an increased risk of colorectal cancer [4,7].

The prevalence and incidence of IBD are increasing with approximately 4.9 million cases worldwide in 2019 [8]. Although not commonly associated with mortality, IBD affects significant morbidity, severely impacting patient quality of life and placing a substantial burden on global economic and healthcare systems already fraught with delivering efficient care and access [8].

Considerable progress in understanding the pathogenesis of IBD has been made in recent years; however, the precise etiology remains unknown. Accumulating evidence suggests that IBD is a heterogeneous disorder of multifactorial etiology involving complex interactions between genetic predisposition, gut microbiota dysbiosis, immune response, and environmental factors, such as cigarette smoking, stress, diet, medications, appendectomy, or pancreatitis [9]. Advances toward understanding IBD etiology have enabled the development of diagnostic techniques and treatment strategies to improve disease severity and patient outcomes [10]. However, further investigation into the disease mechanisms and novel therapies are necessitated to achieve long-term therapeutic success.

### 1.1. ENS and IBD

Substantial bidirectional communication between the enteric neurons and immune cells has been established in both healthy and disease states, emphasizing the significance of ENS in GI immunity [11,12,13,14]. The ENS comprises an interconnected network of enteric neurons and glial cells within the GI tract projecting towards effector structures, such as smooth muscular layers, immune cells, and blood vessels [15]. Crosstalk between the ENS and immune cells results in the production and release of immune and neural mediators, such as cytokines, chemokines, and neurotransmitters [12,16,17].

Previous studies have demonstrated immune infiltration, gross morphological changes, and enteric neuronal loss in tissues from IBD patients and animal models of chronic colitis [18,19,20,21,22,23,24]. Inflammation-induced changes to enteric neuronal structure and function are associated with the development of IBD symptoms, such as diarrhea and/or constipation, heightened sensitivity, and pain, that persist beyond the resolution of active inflammation [25,26,27]. In addition, a compromised GI antioxidant capacity has been associated with inflammation-induced ENS damage [28]. The underlying mechanisms responsible for these effects primarily involve substantial alterations in neurologically regulated processes, encompassing intestinal motility and secretion. While it is widely regarded that ENS abnormalities emerge secondary to inflammation, the occurrence of anomalies in non-inflamed areas of the gut suggests that the ENS may also play a role in IBD pathogenesis [29].

It is evident that the role of neuroimmune interactions in inflammatory conditions is critical for prolonging remission, rendering the ENS an ideal target for the development of novel therapies to remedy the symptoms and underlying pathophysiology of IBD [30].

### 1.2. Current Treatments for IBD

Since the etiology of IBD is yet to be fully elucidated, available therapies are designed to alleviate symptoms, mitigate complications, and control acute mucosal inflammation to prolong remission, rather than target underlying pathological mechanisms [31]. Conventional medical treatments include 5-aminosalicylic acid (5-ASA)-based anti-inflammatories, corticosteroids, immunomodulators, antibiotics, biological therapies using antibodies, and targeting the gut microbiome [31,32].

For patients with mild to moderate UC, 5-ASA-based treatments, i.e., sulfasalazine and mesalazine, are the first line of treatment demonstrated as safe and effective to induce remission and prevent relapse [31]. However, 5-ASAs have been associated with exacerbation of IBD symptoms and serious side effects, such as pancreatitis, pleuritis, myocarditis, and nephritis [33,34].

Corticosteroids have been prescribed for IBD flare-ups for decades due to their broad-spectrum anti-inflammatory capability and are subsequently the recommended treatment for induction of remission in UC patients who do not respond to 5-ASA, as well as in patients with mild to moderate CD [35]. While highly effective for inducing remission, corticosteroids are ineffective for the maintenance of remission due to toxicity effects, contribution to major infection, and a loss of response over time [31,36,37].

Immunomodulators, including thiopurines (azathioprine), are the mainstay treatment for the treatment of moderate to severe CD and active UC where 5-ASA therapy has failed [31]. The complex pathway by which the immunosuppressive anti-metabolite azathioprine exerts its anti-inflammatory effects results in the inhibition of protein synthesis in lymphocytes [38]. Dosage is dependent on the patient’s levels of enzyme serum thiopurine S-methyltransferase (TPMT); when TPMT is low, the risk of myelosuppression, non-melanoma skin cancer, and non-Hodgkin’s lymphoma is enhanced [37]. Thus, the toxicity of thiopurine therapy is highly variable and unpredictable among individuals. While the efficacy of immunomodulators to maintain remission has been established, long-term use is associated with an increased rate of infection and risk of cancer [37,39]. Additionally, the onset of action for anti-metabolites is slow, which is not ideal given these drugs are most effective when the IBD is moderate to severe [40].

The use of biological therapies, such as anti-tumor necrosis factor (TNF) monoclonal antibodies, adalimumab, and infliximab, to induce and maintain remission in IBD patients has increased considerably in past years [31]. Although highly efficacious for moderate to severe UC and CD, most patients either do not respond to initial treatment or lose responsiveness over time [41]. Furthermore, biologic therapies have been associated with reactivating serious infections such as tuberculosis and hepatitis B, and an increased risk for lymphoma and non-melanoma skin cancer [31,37].

Dysbiosis of the gut microbiota has been implicated in the pathogenesis of IBD, leading to the development of strategies targeting microbial composition and modulation [42,43]. The gut microbiome consists of the community of microorganisms within the GI tract, including bacteria, archaea, viruses, and fungi [44]. Microbial composition continuously develops with influences from the environment, diet, age, hygiene, infections, and antibiotic usage [45]. This community of microbiota is the result of a mutually beneficial and harmonious relationship between the microorganisms, often referred to as symbiosis. In turn, the term dysbiosis is used to describe a shift in the balance of microorganisms, interrupting this mutually beneficial relationship, and has been linked to the pathophysiology of IBD [46]. In addition to the dysbiosis, damage to the intestinal epithelial barrier or mucus layer allows bacterial invasion into the lamina propria, promoting the inflammatory response observed in the tissues of IBD patients [44]. Methods for analyzing the gut microbiome in IBD include tissue biopsies, surgical sections, and fecal analysis [47,48,49]. From many of these studies, the changes in microbial composition conflict with one another due to several factors, such as antibiotic usage, age, gender, tobacco smoking, IBD type, analysis method, and GI tract location [49,50,51]. Current treatments that target microbial dysbiosis include probiotics, prebiotics, fecal microbiota transplantation (FMT), dietary interventions, antibiotics, and microbiome profiling. Treatments, such as FMT, constitute ethical and practical challenges, such as donor screening [52]. The response to microbiome-targeting treatments can vary due to the heterogeneity of IBD. Furthermore, the diversity of the microbiome composition among IBD patients adds a further level of complexity when treating the disease [53].

It is evident that the efficiency of current conventional IBD therapies is limited by the severity of side effects, loss of patient responsiveness, long-term toxicity, and/or failure to induce and maintain remission [54]. Therefore, the development of novel IBD therapies targeting pathophysiological mechanisms and pathways that are precise, efficacious, and reduce the occurrence of off-target effects is crucial.

### 1.3. Oxidative Stress and IBD

Oxidative stress has been implicated with the development and progression of IBD, therefore, redox-targeted therapy is a promising option for treatment [55]. Reactive oxygen species (ROS) are mostly generated as by-products of oxidative metabolism during mitochondrial respiration, as well as in cellular response to xenobiotics, cytokines, and bacterial invasion [56]. Aerobic organisms develop a comprehensive endogenous antioxidant defense system to maintain cellular redox homeostasis [57]; oxidative stress refers to the state of imbalance between ROS production and the capacity of the antioxidant defense system to mount an effective response, in favor of oxidants [58]. Modulation of intracellular levels of ROS is vital for cellular homeostasis as different ROS levels produce contrasting biological responses. ROS functions as second messenger, signaling molecules to regulate cellular physiological and biological processes when produced at low to moderate concentrations. However, excessive amounts of ROS overwhelm the antioxidant defense system and result in redox imbalances disrupting cellular integrity and functions, including damage to cellular and mitochondrial lipids, proteins, and DNA [59,60]. Production of ROS by mitochondria is critical as it underlies oxidative damage in many pathologies, including inflammatory, cardiovascular, carcinogenic, autoimmune, and neurological degenerative diseases [60,61].

Previous studies have reported elevated levels of biomarkers for oxidative stress and ROS-mediated damage, together with reduced antioxidant levels in IBD patients and animal models of colitis [62,63,64,65,66]. It is indicated that an oxidative stress-induced exaggerated inflammatory response alters epithelial barrier permeability allowing luminal pathogen invasion and leukocyte infiltration [67]. Therefore, the cumulative damage results in intestinal mucosal necrosis and ulceration associated with IBD [68]. Increased expression and activation of nuclear factor kappa B (NF-κB) has been detected in the colonic mucosa of patients with IBD [69,70]. Under physiological conditions, ROS-activated intracellular signaling pathways, such as NF-κB, are recognized to play a role in the maintenance of the intestinal epithelial barrier function and coordination of the epithelial immune response in microorganisms [71,72]. However, oxidative activation of NF-κB and activator protein-1 (AP-1) signaling stimulates expression of pro-inflammatory cytokines in intestinal epithelial cells, including tumor necrosis factor alpha (TNF-α), interleukin (IL)-1, IL-6, IL-8, and cyclooxygenases-2 (COX-2) [73]. These cytokines are associated with exacerbating existing inflammation and promoting carcinogenesis in IBD [74]. Therefore, given that persistent oxidative stress is considered to play a role in the pathogenesis and progression of IBD, it is indicative that sustained NF-κB signaling stimulated by excessive ROS exaggerates the chronic intestinal inflammation in the mucosa of IBD patients.

In IBD, there is an abnormal immune response against the gut microbiome [75]. The mucosa provides a habitat for the microbiome, and in turn, the microbiome influences the health and function of the mucosa. Thus, there is a dynamic and bidirectional relationship between the mucosa and microbiome [76]. The composition of gut microbial communities is influenced by intestinal oxygenation. Healthy intestines are characterized by low oxygen levels and large bacterial communities of obligate anaerobes, whereas in IBD, chronic inflammation results in increased release of hemoglobin carrying oxygen and reactive oxygen species into the intestinal lumen, creating a microenvironment that favors facultative anaerobes [77,78,79]. The subsequent decrease in obligate anaerobes that release anti-inflammatory compounds initiates an increased inflammatory response, inaugurating a positive feedback loop that accelerates the disease process [79,80,81]. Therefore, oxidative stress and microbial dysbiosis are interconnected; the ROS generated during inflammation can directly compromise the integrity of the epithelial barrier, stimulating an immune response and triggering the production of microbial metabolites.

Oxidative stress and chronic inflammation intertwine as key pathologic factors contributing to enteric neuropathy [82,83]. The effects of oxidative stress on the ENS can be cumulative, affecting structural and functional changes [84]. Previous studies report a significant loss of enteric neurons, but an increased proportion of neuronal nitric oxide synthase (nNOS) neurons in the myenteric plexus in colonic tissues from IBD patients and experimental models of colitis [83,85,86,87,88]. Primarily inhibitory motor neurons, nNOS neurons are responsible for the relaxation of the intestinal smooth muscle cells, and altered chemical coding of enteric neuronal subpopulations is associated with impaired smooth muscle contractility and intestinal dysmotility [89]. Changes in the size and proportion of nNOS neurons have been associated with oxidative stress indicating that sensitivity of colonic tissue to redox imbalance may arise from inadequacy of antioxidant defense systems under pathological conditions [84,90].

Colon biopsies from IBD patients and animal models of colitis have shown increased levels of nitric oxide (NO) [91,92] and 8-hydroxydeoxyguanosine (8-OHdG), a common index for examining oxidative damage to DNA [93,94]. Furthermore, 8-OHdG levels are reported to be permanently elevated in CD patients, independent of disease activity [94]. There is only one study that has evaluated the presence of 8-OHdG in myenteric neurons [19]. Results of this study demonstrated increased levels of 8-OHdG identifying oxidative stress-induced DNA damage within myenteric neurons in colons from *Winnie* mice, a spontaneous model of chronic intestinal inflammation. In *Winnie* mice, a decrease in DNA damage was demonstrated after treatment with APX3330, an apurinic/apyrimidinic endonuclease/redox factor-1 (APE1/Ref-1) redox domain inhibitor [19]. Furthermore, APX3330 ameliorated colonic dysmotility, altered GI transit, and enteric neuropathy, including reducing superoxide production in myenteric neurons and preventing neuronal death, while providing anti-inflammatory and antioxidant effects in *Winnie* mice. Therefore, treatment with APX3330 had prominent therapeutic effects in a preclinical animal model of IBD, providing evidence of a potential novel therapy to protect the ENS from IBD-induced injury.

As there is limited pursuance in free radical antioxidants for the treatment of IBD, new therapeutic strategies should aim to obstruct the major sources of oxidative stress contributing to enteric neuropathy.

## 2. APE1/Ref-1

An essential regulator of cellular response to oxidative stress, APE1/Ref-1 is a dual-functioning protein with major roles in DNA repair and redox signaling [95,96]. The subcellular localization of APE1/Ref-1 is predominantly in the nucleus; however, it is also found in the mitochondria and other areas of the cytoplasm [97,98,99]. APE1/Ref-1 functions as the main apurinic/apyrimidinic endonuclease in the base excision repair (BER) pathway, accounting for 95% of abasic site repairs [100]. The BER pathway repairs DNA damaged by oxidation, deamination, and alkylation to prevent mutagenesis and promote cellular survival. Given the vulnerability of neurons to ROS and altered BER function in neurological disorders; repair of oxidative DNA damage by efficient BER activity is considered protective against neurodegeneration [101].

In its role as a redox signaling protein, APE1/Ref-1 acts via redox-dependent mediation of DNA binding of transcription factors, involved in inflammatory regulation, immune response, angiogenesis, and cell survival [96,102]. Several studies have demonstrated enhanced DNA binding activity for AP-1, hypoxia-inducible factor-1 (HIF-1), NF-kB, p53, and STAT3 to be associated with APE1/Ref-1 [103,104,105,106,107,108]. The capacity of APE1/Ref-1 to maintain transcription factors in an active reduced state subsequently affects the modulation of the expression of genes involved in oxidative stress and maintains genomic stability [96]. Silencing of APE1/Ref-1 via siRNA increases intracellular ROS production [109,110]. Correspondingly, blocking APE1/Ref-1 redox activity reduces the DNA binding activity of AP-1, HIF-1α, and NF-κB [111,112]. Cells are more susceptible to oxidative stress when expression of APE1/Ref-1 is compromised; reduced APE1/Ref-1 is associated with increased cytotoxicity of neurons, impacting cell survival and driving apoptosis [113].

On the other hand, APE1/Ref-1 mediation of fundamental transcription factors that control cell cycle arrest and apoptotic programs can promote the growth, migration, and survival of cancer cells, as well as angiogenesis in the tumor microenvironment [95]. Overexpression in cancer cells and irregular cytoplasmic versus nuclear distribution of APE1/Ref-1 is associated with tumor aggressiveness and poorer patient prognosis in many tumor types, including colorectal cancer [114,115].

The dual functions of APE1/Ref-1 are molecularly distinct and completely independent in their function [116]. The apurinic/apyrimidinic endonuclease activity of APE1/Ref-1 is critical for maintaining genomic stability and cellular existence, evidenced by the embryonic lethality of murine APE1/Ref-1 knockout models and the inability to establish viable cell lines completely deficient for APE1 [96]. APE1/Ref-1 redox function is significant for cellular response to oxidative stress, however, the implications of cancer development and progression have motivated recent investigations into inhibition of the redox activity of APE/Ref-1 [19,96,112]. Compounds that specifically inhibit the redox function of APE1/Ref-1 do not obstruct DNA repair activity and previous studies suggest that APE1/Ref-1 redox inhibitors actually augment the endonuclease repair activity of APE1/Ref-1 in the hippocampus, sensory, and enteric neurons, thus providing a neuroprotective effect [19,117,118].

### APE1/Ref-1 in Intestinal Inflammation

Oxidative stress is implicated in inflammatory conditions resulting in altered APE1/Ref-1 response; without intercept of redox activation, cell apoptosis and carcinogenesis are likely [119]. Elevated APE1/Ref-1 expression has been reported in tissues from IBD patients with active inflammation and in an animal model of colitis, signifying underlying oxidative stress within the gut [62,63,120]. Furthermore, increased expression of APE1/Ref-1 in colon sections from UC patients has been associated with an enhanced pro-inflammatory response, which is a precursor for colorectal cancer (CRC) susceptibility [63,120].

In intestinal epithelial cells, APE1/Ref-1 plays a major role in controlling the onset of oxidative stress-based inflammatory processes by modulating NF-κB-mediated IL-8 gene expression [121]. Studies have shown enhanced binding and IL-8 promoter activity in colitis and cancer where APE1/Ref-1 is overexpressed [120,122]. Silencing APE1/Ref-1 in gastric epithelial cells infected with *Helicobacter Pylori* inhibited IL-8 expression [123]. These findings illustrate the importance of APE1/Ref-1 in IL-8 regulation.

In addition to IL-8, IL-6 also has a crucial role in the development and progression of uncontrolled intestinal inflammatory processes characteristic of IBD [124]. Secretory APE/Ref-1 induces IL-6 expression in response to inflammatory challenges, such as lipopolysaccharides (LPS) and TNF-α stimulation, and increased IL-6 expression enhances excessive APE1/Ref-1 secretion in a feedforward loop [125]. Furthermore, silencing of APE1 expression by siRNA decreases the release of IL-6 [126]. Expression and activation of STAT3 are increased in intestinal epithelial cells during active IBD [127]. Given that IL-6 activates STAT3 and constitutively activated STAT3 augments APE1/Ref-1 expression, an underlying role of APE1/Ref-1 in the pathogenesis of GI inflammation is indicated [128].

Increased expression of APE1 in IBD is positively correlated with microsatellite instability (MSI) [63]. In chronic inflammatory diseases such as IBD, MSI is affected by the overproduction of free radicals saturating the ability of the cell to repair DNA damage preceding replication [129,130]. Prolonged intracellular stress induced by chronic inflammation and associated increases in APE1 lead to cumulative genomic instability as the endonuclease activity of APE1 produces cytotoxic DNA repair intermediates [131]. It is, therefore, considered that chronic inflammation in IBD causes adaptive increases in APE1 levels, which paradoxically enhance MSI in affected tissues, and enzymes that normally facilitate DNA repair are instead causing mutations [132]. Chromosomal instabilities have been detected in dysplastic tissues from patients in the early stages of IBD [133]. These instabilities contribute to gene mutations leading to colon carcinogenesis and a 20–30-fold increased risk of developing colorectal cancer (CRC) [134].

Despite considerable evidence supporting enteric dysregulation in IBD, the effect of oxidative stress and the mechanistic role of APE1/Ref-1 within the ENS has not fully been elucidated [83,86,87,88,89]. Specific inhibition of APE1/Ref-1 redox function has shown promising effects in preventing enteric neuropathy and alleviating intestinal inflammation in a murine model of spontaneous chronic colitis [19]. Hence, the APE1/Ref-1 redox domain inhibitor APX3330 provides an opportunity to target specific redox mechanisms of the oxidative stress response associated with intestinal inflammation and provide further understanding of its role in IBD pathogenesis.

## 3. Targeting APE1/Ref-1 as a Therapeutic Approach

Currently branded as APX3330, (E)-3-(5,6-dimethoxy-3-methyl-1,4-dioxocyclohexa-2,5-dienyl)-2-nonylpropenoic acid (E3330) has been synthesized as a known small-molecular APE1/Ref-1 redox function inhibitor [135]. APX3330 has shown efficacy and safety in clinical trials for oncology and diabetic retinopathy [136,137]. The use of APX3330 facilitates endeavors in further understanding of APE1/Ref-1 redox interactions and isolation, providing an opportunity for specific therapeutics targeting pathophysiological mechanisms of IBD. APX3330 is a highly selective inhibitor of Ref-1 redox activity specifically binding to sites located on the APE1/Ref-1 redox domain while leaving the DNA repair domain untouched [138,139,140,141]. The specific inhibition activity of APX3330 blocks Ref-1’s ability to convert transcription factors from their oxidized, inactive state to an active, reduced state. APX3330 has been shown to suppress the production of pro-inflammatory cytokines and inflammatory mediators in murine macrophages resulting in the inability of NF-kB and AP-1 to bind to their target DNA sequence [122]. The inhibition activity of APX3330 downregulates secretion of the inflammatory cytokines IL-6 and IL-12, and the inflammatory mediators PGE2 and NO, as well as expression of COX-2 and iNOS which regulate the production of PGE2 and NO [122]. Moreover, APX3330 has been demonstrated to inhibit HIF-1 DNA binding activity under the same mechanisms that HIF-1 regulates APE1/Ref-1 expression; although it is itself regulated by APE1/Ref-1 [140,142,143]. Inhibition of HIF-1 and NFkB function in patients treated with APX3330 resulted in the suppression of genes downstream of these transcription factors in a cancer clinical trial [144]. In a murine model of spontaneous chronic colitis, APX3330 affected repair of inflammation-induced ENS damage, and ameliorated IBD symptoms, in addition to demonstrated immunomodulatory function [19]. Thus, targeting the specific inhibition of APE1/Ref-1 redox pathways while preserving the DNA repair pathway in intestinal inflammation is promising and a potential novel treatment for IBD and its associated enteric neuropathy (Figure 1).

## 4. Conclusions and Future Directions

Previous studies have demonstrated that APE1/Ref-1 simultaneously affects inflammation and oxidative stress. Although the effects of APE1/Ref-1 on diverse transcriptional targets have been extensively examined in various types of cancers and diseases, its impact on other transcriptional targets within the context of IBD remains unexplored. In addition, investigations elucidating the role of APE1/Ref-1 in inflammation-induced enteric neuropathy are warranted. Furthermore, the overall range of effects of APE1/Ref-1 redox inhibitors in IBD is currently unknown. Thus, further endeavors are imperative to comprehend the mechanism of APE1/Ref-1 and its redox inhibitors in chronic intestinal inflammation.

APE1/Ref-1 redox inhibitors provide a novel therapeutic strategy for managing colitis compared to conventional treatments. They present distinct advantages over current drugs for treating colitis by not only targeting inflammation and oxidative stress but also repairing DNA damage and are oral agents. The benefit of APE1/Ref-1 inhibitors as a prospective remedy for IBD lies in their minimal systemic adverse effects reported in clinical trials in cancer and diabetic retinopathy patients. Given the significance of the redox-regulated transcriptional control of APE1/Ref-1 in intestinal inflammation, targeting APE1/Ref-1 with APE1/Ref-1 redox inhibitors offers a novel treatment approach for IBD that has the potential to circumvent the challenges associated with current therapies and improve disease outcomes in many patients.

## Figures and Tables

**Figure 1 biomolecules-13-01569-f001:**
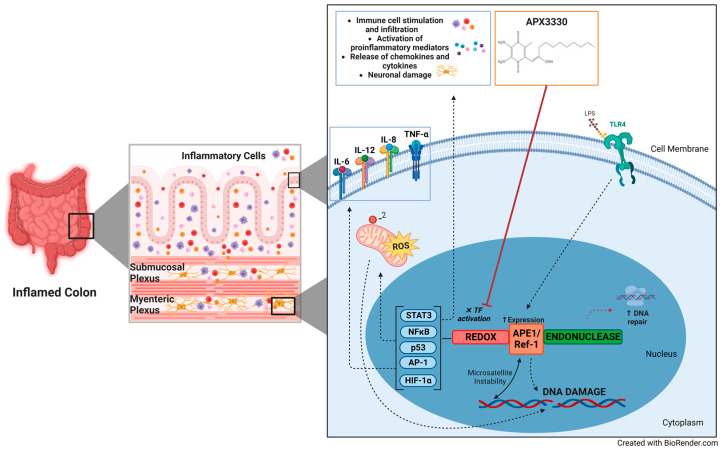
APE1/Ref-1 role in inflammatory bowel disease and the potential of redox signaling inhibition. The activation of transcription factors through the redox function of APE1 is implicated in the promotion of inflammation in inflammatory bowel disease. Furthermore, the subsequent activation of downstream mediators directly affects the inflammatory responses. By inhibiting the redox signaling pathway using APX compounds, the excessive production of pro-inflammatory cytokines, chemokines, reactive oxygen species (ROS), and neuronal damage can be reduced, leading to the alleviation of inflammation and gastrointestinal functions. Additionally, the repair function of APE1 plays a crucial role in mending the DNA or RNA lesions caused by inflammation and oxidative stress. In summary, the utilization of APX compounds can effectively diminish inflammatory markers, ROS levels, and enteric neuropathy, while also enhancing the DNA repair function, thereby presenting a novel therapeutic opportunity for IBD (created with BioRender.com (accessed on 30 September 2023)). Abbreviations: TFs: transcription factors; IL: interleukin; TNF-α: tumor necrosis factor-alpha; ROS: reactive oxygen species; STAT3: signal transducer and activator of transcription 3; NFκB: nuclear factor kappa-light chain enhancer of activated B cells; AP-1: activator protein 1; HIF1α: hypoxia-inducible factor 1 subunit alpha; redox: reduction-oxidation; LPS: lipopolysaccharide; TLR4: toll-like receptor 4; DNA: deoxyribonucleic acid.

## Data Availability

No new data were created.

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
