# Peer review of "APE1/Ref-1 as a Therapeutic Target for Inflammatory Bowel Disease"

_biomolecules, 2023, doi:10.3390/biom13111569_

Round 1
Reviewer 1 Report
Inflammatory bowel disease (IBD) is a chronic condition characterized by chronic inflammation of the gastrointestinal tract. The prevalence of IBD is increasing globally, necessitating the development of novel therapeutic approaches due to the limitations of current treatments. Recent research has identified apurinic/apyrimidinic endonuclease 1/reduction-oxidation factor 1 (APE1/Ref-1) as a potential target in inflammatory diseases. APE1/Ref-1 regulates crucial transcription factors involved in significant pathways, making it an attractive candidate for IBD therapy. This review provides an overview of conventional IBD treatments, discusses the role of APE1/Ref-1 in intestinal inflammation, and explores the potential of a small molecule inhibitor targeting APE1/Ref-1 redox activity to modulate inflammation, oxidative stress response, and enteric neuronal damage in IBD. But I have several following concerns:
1. The structure of this manuscript is also somewhat odd for a review article. The first part should be "Introduction".
2. In Lines 32 and 33, "Inflammatory Bowel Disease" should be "Inflammatory bowel disease", "Ulcerative Colitis" should be "ulcerative colitis", "Crohn’s Disease" should be "Crohn’s disease".
3. Abbreviations should be defined When thery appears in the first time. Such as "COX-2" in Line 140.
4. If Figures are a copy of the reported Figures or made with software, please apply for copyright or indicate the quotation.
5. In Line 310, "comprehending" should be "comprehend"
6. In the Section 3, there is only one secondary title, which is suggested to be removed.
7. In the "Conclusion and Future Directions" section, the authors should add a discussion of the advantages of APE/Ref-1 inhibitors over current drugs for the treatment of colitis. In addition, what are the dilemmas and possible solutions in the current development of APE/Ref-1 targeted drugs?
8. Please unify the format of references in the article, including the author's name, the case of words in the title of the article, the writing of the name of the journal, and the page number.
Moderate editing of English language required
Author Response
Reviewer 1
The manuscript submitted by Sahakian, and cols is a comprehensive review exploring the putative beneficial effects associated with targeting the APE1/Ref-1 pathway in IBD. Below, I have detailed all concerns regarding this review:
The authors describe different pharmacological approaches, including its mechanisms, that are frequently used to treat IBD. However, they have not mentioned those approaches targeting the microbial composition/modulation. Thus, considering the direct/indirect effects of microbial modulation towards the IBD onset and outcome, it is of utmost importance to describe the main strategies used in this regard.
Response: Thank you for your comment. We have inserted a paragraph on targeting microbial composition and modulation; please see lines 123-144.
Authors have described the relationship between Oxidative Stress and IBD, however, they forgot to add information concerning the impact of oxidative stress towards gut microbiota modulation. It is of common knowledge that in IBD the combination between inflammation and intestinal dysbiosis improves pO2 thus, reducing the local availability of anaerobic microbes, which in turns favours aerobic bacteria growth. Hence, they must describe and explore the impact of therapeutic candidates in modulating such aspect in IBD.
Response: Thank you for your comment. We have inserted a paragraph on the mucosa and microbiome alterations due to oxidative stress. Please see lines 186-199.

Reviewer 2 Report
The manuscript submitted by Sahakian and cols is a comprehensive review exploring the putative beneficial effects associated with targeting the APE1/Ref-1 pathway in IBD. Bellow, I have detailed all concerns regarding this review:
- The authors describe different pharmacological approaches, including its mechanisms, that are frequently used to treat IBD. However, they have not mentioned those approaches targeting the microbial composition/modulation. Thus, considering the direct/indirect effects of microbial modulation towards the IBD onset and outcome, it is of utmost importance to describe the main strategies used in this regard.
- Authors have described the relationship between Oxidative Stress and IBD, however, they forgot to add information concerning the impact of oxidative stress towards gut microbiota modulation. It is of common knowledge that in IBD the combination between inflammation and intestinal dysbiosis improves pO2 thus, reducing the local availability of anaerobic microbes, which in turns favours aerobic bacteria growth. Hence, they must describe and explore the impact of therapeutic candidates in modulating such aspect in IBD
Author Response
Reviewer 2
Inflammatory bowel disease (IBD) is a chronic condition characterized by chronic inflammation of the gastrointestinal tract. The prevalence of IBD is increasing globally, necessitating the development of novel therapeutic approaches due to the limitations of current treatments. Recent research has identified apurinic/apyrimidinic endonuclease 1/reduction-oxidation factor 1 (APE1/Ref-1) as a potential target in inflammatory diseases. APE1/Ref-1 regulates crucial transcription factors involved in significant pathways, making it an attractive candidate for IBD therapy. This review provides an overview of conventional IBD treatments, discusses the role of APE1/Ref-1 in intestinal inflammation, and explores the potential of a small molecule inhibitor targeting APE1/Ref-1 redox activity to modulate inflammation, oxidative stress response, and enteric neuronal damage in IBD. But I have several following concerns:
The structure of this manuscript is also somewhat odd for a review article. The first part should be "Introduction".
Response: Line 31 – changed “Inflammatory Bowel Disease” to “Introduction”
In Lines 32 and 33, "Inflammatory Bowel Disease" should be "Inflammatory bowel disease", "Ulcerative Colitis" should be "ulcerative colitis", "Crohn’s Disease" should be "Crohn’s disease".
Response: Changes made in text, please see lines 33.
Abbreviations should be defined When they appear in the first time. Such as "COX-2" in Line 140.
Response: COX-2 abbreviation defined, please line 181.
If Figures are a copy of the reported Figures or made with software, please apply for copyright, or indicate the quotation.
Response: (Created with BioRender.com) inserted under Figure as per requirements of the software copyright. Please see line 352.
In Line 310, "comprehending" should be "comprehend”.
Response: “Comprehending” changed to “comprehend”, please see line 366-367.
In the Section 3, there is only one secondary title, which is suggested to be removed.
Response: Secondary title is removed.
In the "Conclusion and Future Directions" section, the authors should add a discussion of the advantages of APE/Ref-1 inhibitors over current drugs for the treatment of colitis. In addition, what are the dilemmas and possible solutions in the current development of APE/Ref-1 targeted drugs?
Response: Paragraph added on the topic, please see lines 369-378.
Please unify the format of references in the article, including the author's name, the case of words in the title of the article, the writing of the name of the journal, and the page number.
Response: Thank you for your comment; we have made changes to the references to unify formatting according to Reviewer’s suggestion.

Round 2
Reviewer 1 Report
The authors have addressed all my concerns. I recommend accepting this manuscript in current form.